# Impact of Nutritional Supplementation on Canine Dermatological Disorders

**DOI:** 10.3390/vetsci7020038

**Published:** 2020-04-03

**Authors:** Andrea Marchegiani, Alessandro Fruganti, Andrea Spaterna, Elena Dalle Vedove, Benedetta Bachetti, Marcella Massimini, Francesco Di Pierro, Alessandra Gavazza, Matteo Cerquetella

**Affiliations:** 1School of Biosciences and Veterinary Medicine, University of Camerino, 62024 Matelica, Italy; alessandro.fruganti@unicam.it (A.F.); andrea.spaterna@unicam.it (A.S.); alessandra.gavazza@unicam.it (A.G.); matteo.cerquetella@unicam.it (M.C.); 2Research and Development Unit (NIL), C.I.A.M. srl, 63100 Ascoli Piceno, Italy; elena@nilitaly.com (E.D.V.); benedetta@nilitaly.com (B.B.); marcella@nilitaly.com (M.M.); 3Scientific Dept, Velleja Research Institute, 20125 Milan, Italy; f.dipierro@vellejaresearch.com

**Keywords:** complementary feed, canine, skin

## Abstract

Nutritional supplements, also known as complementary feeds, are products administered with the aim of furnishing health benefits, regardless of nutritional needs. They have been used since ancient times in veterinary dermatology, and a number of studies have focused on investigating the health benefits of some ingredients found in commercially available complementary feed for dogs. The aim of this paper is to review the literature available on the use of nutritional supplementation for the management of canine skin diseases, critically appraising the clinical efficacy of such interventions and summarizing the current state of knowledge. This review highlights how these feeds can be considered useful in the management of dermatological disorders and outlines their beneficial effects in the prevention of dietary deficiencies and treatment of diseases, alone, or in addition to conventional pharmacological therapy. In recent years, nutritional supplements have found increasing potential application in veterinary medicine, and the scientific proofs of their beneficial effects are described in this review.

## 1. Introduction

The skin is the largest organ in the bodies of dogs and is perpetually exposed to a range of different distressing internal and external factors [1]. Moreover, skin is one of the most important immunologic structures of the body, with relatively high physiological requirements in terms of nutrients [1]. In view of this, any subtle changes in the nourishment of skin, as well as variations with respect to its immunological state, can have marked effects on skin and coat condition [2,3].

Dermatological disorders represent the most commonly presented complaints in veterinary practice, accounting for almost a quarter of all clinical work [4]. Canine cutaneous problems refer to skin abnormalities that are usually caused by inadequate nutrition and/or hormonal imbalances, in addition to numerous agents, including microorganisms, physical or chemical agents, and immunological reactions [1,5]. Among these conditions, flea infestations, bacterial infections, and allergic skin diseases, such as canine atopic dermatitis and neoplasia, are the most frequently encountered [6,7]. Furthermore, genetic and environmental factors play important roles in the development of these diseases, most of which are inflammatory in nature [8].

Due to the variability of the underlying causes, management of canine dermatological conditions requires a complex approach, usually combining different forms of interventions, such as allergen avoidance and/or specific immunotherapy, antimicrobial, and anti-inflammatory pharmacotherapy [9,10]. Alongside these traditional pharmacological therapies, in the last few years, non-pharmacological approaches have been put forward in the hope of reducing clinical signs of certain skin disorders, especially canine atopic dermatitis [11,12,13]; these have included the use of different natural agents and/or complementary feeds (Table 1).

The North American Veterinary Nutraceutical Council defines a veterinary nutraceutical as “a substance which is produced in a purified or extracted form and administered orally to patients to provide agents for normal body structure and function and administered with the intent of improving the health and well-being of animals” [3]; however, there is no common consensus as yet on the usage of this term. Components often consist of different bioactive compounds acting on multiple targets and pathways [11,14,15]. Consequently, these products, which are marketed as complementary feeds (CF), can potentially interact with multiple different systems in the body under different conditions, including dermatopathies [16]. The strategy of nutritional supplementation has been used since ancient times, and some studies have focused on the investigation of the health benefits of different molecules; despite this, only a small number of appropriately designed trials (randomized, double-blinded, controlled, etc.) can be found in the literature [11]. 

The aim of this paper is to succinctly review recent progress with respect to nutritional supplementation in the management of canine skin diseases, critically appraising the clinical efficacy of such interventions and summarizing the current state of knowledge. 

## 2. Nutritional Supplementations in Veterinary Dermatology

### 2.1. Bioactive Lipids

Aliamides are naturally occurring bioactive lipid compounds that are produced in response to stress and tissue damage, both of which play a central role in the regulation of cutaneous inflammation and immunity [17,18]. Palmitoylethanolamide (PEA) is the parental molecule of the aliamide family and a naturally occurring endocannabinoid-like mediator [19]. Although the mechanisms of action of PEA has not been completely elucidated, it may comprise downregulation of mast cell (MC) degranulation, as well as possible involvement in the process called Autacoid Local Injury Antagonism [19,20,21,22]. PEA levels increase in inflamed tissues, particularly in the skin [23]. A body of research suggests that this type of response is either able to reduce the severity of clinical signs and symptoms or counteract disease progression, supporting the hypothesis of the autoprotective role of PEA [23,24,25]. In fact, it was demonstrated that atopic dogs have skin levels of PEA up to 30-fold higher than healthy dogs [24]. 

In vitro studies have demonstrated that histamine, prostaglandin D2 (PGD2), and TNF-α release were significantly inhibited by the presence of PEA in a dose-dependently manner [22]. In addition, topical application of 2% Adelmidrol gel, another aliamide compound different from PEA, has been shown to significantly increase the granular density of skin MC in cutaneous canine wound biopsies [26].

In veterinary dermatology practice, aliamides provided promising results in the treatment of allergic and inflammatory spontaneous pathologies, especially when MCs are involved [27]. In a randomized, double-blinded study conducted on dogs affected by dust mite hypersensitivity, PEA was orally administered at a dose of 15 mg/kg body weight for 7 days, and it was able to delay the recurrence of the symptoms after allergen exposure [28]. Aliamides were also tested in *Ascaris suum* spontaneous hypersensitivity. A single oral treatment of PEA at three different dosages (3, 10, 30 mg/kg body weight) was able to reduce, significantly and rapidly, the clinical manifestation of hypersensitivity principally at a dose of 10 mg/kg. The same study also provided additional insight concerning the pharmacokinetics of PEA, which is quickly absorbed in the stomach and reaches its maximum plasmatic concentration between the first and second hour, decreasing to the basal level within 4 h [29]. In another randomized, double-blinded vs. placebo study on canine atopic dermatitis (AD), the oral administration of 15 mg/kg/day of PEA was responsible for a statistically significant reduction in Canine Atopic Dermatitis Extent and Severity Index (CADES)-04 scores [30]. 

A multicenter clinical trial demonstrated the efficacy of daily topically administered Adelmidrol 2% in reducing pruritus and skin lesions by 55% and 41%, respectively, in twenty client-owned dogs affected by chronic pruritus of unknown causes [31].

The efficacy and safety of ultra-micronized-PEA (um-PEA) was assessed in an 8-week, open-label, multicenter trial. The objective was to determine if a two-month period of PEA administration could affect pruritus, skin lesions, and quality of life in client-owned dogs affected by symptomatic and stable, non-seasonal canine AD. This study involved 39 veterinary clinics throughout Italy and showed that um-PEA was able to significantly reduce pruritus and skin lesions and improve quality of life (QoL) in nearly 80% of dogs suffering from moderate AD within a month of supplementation. Regarding QoL, dogs regained the normal habits they had before AD as well as their normal playing, working activity, and social interactions [32].

### 2.2. Polyunsaturated Fatty Acids

Omega 3 fatty acids are supposed to produce their beneficial effects by shifting the arachidonic acid (AA) cascade towards production of less inflammatory mediators as prostaglandins and leukotrienes [33]. Although supplementation of n-3 polyunsaturated fatty acids (PUFAs) is commonly recommended for a variety of pruritic and inflammatory diseases to reduce their clinical manifestation [34], it is still to be clarified whether they can be effective as therapeutic agents for pruritus.

The n-6 fatty linoleic acid (LA) is essential for the epidermal barrier function and diets implemented with LA have resulted in a significant decrease in trans-epidermal water loss. The essential fatty acids (EFAs) from n-6 (gammalinolenic acid (GLA)) and n-3 (eicosapentaenoic acid (EPA) and docosahexaenoic acid (DHA)) families have been proven to have anti-inflammatory effects and immunomodulating properties on skin [35,36]. They modulate the eicosanoid production by competing with arachidonic acid (AA) resulting in a shift from pro-inflammatory molecules toward production of leukotriens with anti-inflammatory properties, in addition to decreasing the production of pro-inflammatory cytokines [37]. Although natural anti-inflammatory agents are perceived as side-effect free [38], PUFAs have been shown to have potential side effects, including impairment of platelet and immune function, possible nutrient-drug interactions, gastrointestinal adverse effects, negative consequences on wound healing, and weight gain [39]. 

Papers aimed to determine the efficacy of essential fatty acids in controlling clinical manifestation of canine dermatitis are copious [34,40,41]; despite this, endpoints set by these studies are variable and there is a lack of specific markers.

Sources of omega 3 fatty acids are represented by flax seed and fish oils while omega 6 fatty acids can be found in sunflower oil, borage seed oil, and black currant seed oil. These sources of fatty acids have been tested in different studies for their effects on skin and hair conditions [33,38,42]. Noteworthy, omega-6 fatty acids may have a possible steroid sparing effect in canine AD [42]. No agreement has been reached yet on the optimum ratio and dosage of omega-6 and omega-3 (n-3/n-6) fatty acids in the management of pruritus [34,43,44].

Fatty acids have been widely studied in canine AD. Their supplementation causes different responses when given at different stages of AD, with better clinical outcomes in early stages of it. Topical application of PUFAs in atopic dogs has been shown to improve both CADESI-03 and pruritus scores [45,46].

In addition, PUFAs are able to significantly reduce the dosage of cyclosporine in atopic dogs with no significant changes in CADESI-03 score and QoL [47]. 

### 2.3. Botanical Extracts

Medicinal plants have been used since ancient times for the management of dermatological disorders in both humans and animals due to inflammatory pathway inhibition, epidermal barrier enhancing properties, and their effect against a broad spectrum of bacteria involved in canine dermatological disease [48]. As a result, the target of systemic plant extract-based treatments is represented by inflammatory system modulation while topic treatments are directed mainly toward the restoration of epithelial barrier integrity in addition to support the healing process via antimicrobial activity [48]. 

The main mechanisms of action have been shown to be the interaction with canine immune system and the enhancement of anti-inflammatory drug activity. Thus, plant extract can be used alone or in combination with drugs to reduce their required dose and lessen the adverse effects, in addition to prevent disease or decrease their clinical manifestations [49]. 

Numerous reports in literature investigated the anti-inflammatory effects of *Actinidia arguta* (hardy kiwi) [50,51]. Chinese herbs, such as *Rhemannia glutinosa*, *Paeonia lactiflora,* and *Glycyrrhiza uralensis* have a significant steroid sparing effect when administered in combination with methylprednisolone for the management of canine AD, in addition to having inflammatory, antioxidant, and antimicrobial properties [52,53,54,55,56,57,58,59,60,61]. *Arctium lappa*, *Althaea officinalis*, *Malva sylvestris*, and *Ribes nigrum* also are attributed with antioxidant and anti-inflammatory activities [62].

In veterinary medicine, few works showed an in vitro effect of plant extract on the inflammatory pathway in canine cell lines. Santoro and collaborators demonstrated the inhibition ability of Boldo and Meadowsweet plant extracts on the expression of antimicrobial peptides and inflammatory markers in canine keratinocytes [63]. Di Cerbo and collaborators evaluated the reduction of IFNϒ in canine peripheral blood mononuclear cell (PBMC) and T lymphocytes after treatment with *Ascophyllum nodosum*, *Cucumis melo*, *Haematococcus pluvialis*, *Curcuma longa*, *Camellia sinensis*, *Punica granatum*, *Polygonum cuspidatum*, *Echinacea purpurea*, *Grifola frondosa*, and *Glycine max* [64].

In humans, skin disorders caused by an excess of sebum, seborrheic skin, and acne have also been addressed using the *Serenoa repens* (saw palmetto) fruits extract. Although this compound is endowed with only a weak capability to inhibit 5-alpha-reductase (an enzyme also involved in sebum production), it has been shown to reduce the oily appearance of the skin [65]. In dogs, the same extract has been used for the management of cutaneous manifestation of benign prostatic disease, showing an ameliorative effect on seborrhea as well as demonstrating its safety profile in dogs [66]. These last data could push to verify the possible use of the saw palmetto extract in the seborrheic skin of the dog. 

In the last decade, the number of publications with in vivo and in vitro studies about plant extract effects has tremendously increased, highlighting the importance of considering the type of extract and the extraction method used [67]. The amount of active constituents in the plant material, in fact, can be influenced by several environmental factors, but also by the plant parts and the subspecies used [48], while the bioavailability rate can be influenced by the presence in the botanical extract of natural carriers and inhibitors of P-glycoprotein and glucuronidation enzymes [68]. 

In addition to orally administered compounds, many botanical extracts have been proven to have anti-inflammatory, antibacterial, and antifungal effects when applied topically. These compounds have been successfully used in the management of different dermatological conditions, such as canine atopic dermatitis, pruritic dermatitis, pyoderma, otitis externa, infected wounds, and dermatophytosis, making them an interesting option for the topical treatment of dermatological conditions [48,69].

### 2.4. Probiotics

One of the most important functions of the gut microbiota is to be an integral part of the intestinal barrier, which defends the host from pathogens and modulates the immune system. In addition, it plays an essential role in preserving structural and functional integrity of the gut itself [70,71,72].

In this regard, the gut and skin share the role of primary interface with the external environment [70]. Many studies have shown a deep and bidirectional connection between the gut and skin, and numerous data link gastrointestinal homeostasis to skin health [73]. It is known that gastrointestinal (GI) problems are often accompanied by cutaneous manifestations, as well as the GI system, especially the gut microbiome, and appears to participate in the pathophysiology of many inflammatory diseases [70]. Nevertheless, the exact mechanisms by which the gut exerts its cutaneous effects are only beginning to be revealed and thought to be mostly due to the modulation of the immune system and variation of the microbiome [73].

Many skin diseases, such as AD, can be considered as a potential manifestation of a systemic disorder related to gut dysbiosis and increased intestinal permeability, which may arise even in the absence of GI clinical signs [73]. Modulation of the canine intestinal microbiota is getting more and more attention as a way for attenuating atopic dermatitis and other skin problems, as part of a multimodal approach [9]. In contrast to the traditional monotherapy mode of treatment, the last updates from the International Task Force for Canine Atopic Dermatitis (CAD) highlight the concept of multimodal management techniques that involve different targets along the pathway [9].

While most people and pets are first given probiotics for digestive-related reasons, recent human and animal studies have shown promises in the prevention and treatment of dermatological diseases [74,75,76]. Probiotic treatments may be considered as an integral part of the multimodal approach that may help in the long-term efficient management of AD.

Several studies have been conducted to assess the impact of probiotics in both prevention and treatment of dogs with AD. Among probiotics, *Lactobacillus* and *Enterococcus* are the most studied taxa for dermal purposes. The exact mechanism of action of probiotics on skin disorders has yet to be fully elucidated; the supposed mechanism is the modulation of the immune response toward a Th1 cell-mediated response [77].

Marsella and collaborators found best evidence for probiotic supplementation in bitches and puppies using *Lactobacillus rhamnosus* strain GG (LGG) in long-term prevention in a validated experimental model of AD [78,79]. Although LGG is a human strain, it has the ability to survive to gastrointestinal transit in dogs and it was selected due to the promising results shown in children [80]. Moreover, the reduction of the dose and/or frequency of drugs required to manage AD to relieve their associated side effects, can be also considered a significative outcome. A novel treatment procedure for CAD should comprise long-term strategies to combat aggravating factors and reduce the use of drugs. In this regard, Ohshima-Terada and collaborators evaluated the efficacy of oral administration of *Lactobacillus paracasei* K71 (K71) in canine AD [81]. K71 was selected upon the promising results reported in mice, humans, and dogs [82,83,84]. Supplemented dogs exhibited an improvement of the CADESI, Visual Analog Scale (VAS), and pruritus scores when compared with their baselines. The mechanisms by which K71 may potentially influence the atopic inflammatory cascade have still to be fully elucidated. One of the most plausible hypotheses is that K71 downregulates IgE production via the induction of T helper 1 cytokine milieu, similar to other lactic acid-producing bacteria [81]. Such effect was also observed in ovalbumin-immunized Balb/c mice in which the oral administration of K71 reduced total and allergen-specific IgE levels and induced interleukin-12 and interferon-c production, suppressing interleukin-4 production [82].

In another study, *Lactobacillus sakei* supplementation resulted in significant difference in the Pruritus Visual Analogue Scale (PVAS) score in experimental dogs [85]. 

A body of research is directed toward the evaluation of heat-killed bacteria registered strains. The hypothesized mechanisms by which these bacteria may potentially influence AD symptoms is the modulation of the gastrointestinal microbiome, even if it is not fully elucidated [86,87]. Moreover, there are no data on the possible presence of a cutaneous-intestinal microflora axis and if this can be mutually influenced. 

### 2.5. Vitamins and Minerals

Changes in skin and hair coat may be due to many kinds of nutritional deficiencies. Good nutrition is essential to preserve normal skin health, and this may explain why nutritional supplements are commonly used in skin and hair care. Normal keratinization requires an adequate supply of several nutrients and micronutrients, including vitamins and minerals [88].

Deficiencies in numerous essential amino acids, fatty acids, vitamins, and minerals can cause many types of modifications in skin structure or function. Epidermal atrophy may be seen with protein, calories, and vitamin deficiencies. 

Magnesium, zinc, pantothenic acid, pyridoxine, biotin, vitamin A, or essential fatty acid deficiencies may cause skin disease histologically characterized by hyperkeratosis, parakeratosis, and acanthosis as have been described in a different range of diseases, such as distemper, pemphigus foliaceus, discoid lupus erythematosus, zinc-responsive dermatosis, ichthyosis, and necrolytic migratory erythema [89]. Pigment changes can occur with deficiencies of copper, cysteine, or pantothenic acid. Alopecia or changes in the sebaceous glands may arise from zinc, biotin, or riboflavin deficiencies. Riboflavin deficiency can cause a dry flaky dermatitis with reddening of the skin and hair loss. Biotin deficiency can cause the hair to become thin or loose pigment and the skin to become dry and greasy [90]. Along with these direct effects, suboptimal nutrition may enhance susceptibility to parasites, such as mange mites, fleas and lice, as well as increased susceptibility to skin infections [90,91,92]. 

Such deficiencies are rare among pets fed with quality commercial pet foods; however, since table scraps, treats, and other imbalanced foods can constitute a large portion of the diet for many pets, deficiencies should be taken into account. Moreover, certain animals may have genetic or metabolic differences that may respond to intakes greater than those considered adequate to avoid dietary deficiencies. In the promising field of nutritional supplementation, vitamin and mineral supplements can help provide optimum nutrient requirements and cope with many dermatologic signs associated with dietary deficiencies. Vitamin and mineral supplements can be useful to fulfil nutritional requirements of healthy pets; moreover, they may be beneficial as adjunctive treatments for the management of dermatological diseases. Most of the scientific evidence supports the use of vitamin A, E, D, and zinc as effective complementary treatment.

Vitamin A has been successfully used to treat seborrhea in American cocker spaniels, while topical treatments have been used for several skin conditions in various breeds, as well as it was shown to be beneficial in patients with marked follicular plugging and hyperkeratosis [93,94].

Traditionally, chronic skin disorders have been cured mainly with traditional medicine, especially glucocorticoids and immunosuppressant drugs; only recently, researchers have turned their attention on complementary treatments. In this context, nutritional supplements can be beneficial for those dogs that poorly tolerate long-term use of drugs and can often reduce the use of drugs itself. 

Vitamins D supplementation has shown to be beneficial in case of AD [95] while vitamin E acts as an antioxidant improving the clinical status of dogs with moderate AD [96,97,98], reducing the biosynthesis of prostaglandin E2 [99], inhibiting formation of IgE and decreasing its amount [100,101].

Among minerals, zinc is considered critical for the normal skin development and deficiency leads to parakeratosis and dermatitis. Rapidly dividing cells, such as those of the epidermis, are particularly dependent on zinc [88]. The consequences of zinc deficiency on the skin—from reduced intake, absorption, or metabolism—have been described in dogs, and include impaired wound healing, erythema, alopecia, crusting, and scaling [88].

Zinc-responsive dermatitis is generally reported in puppies or zinc-sensitive breeds (primarily Siberian Huskies and Alaskan Malamutes) although it can occur in other breeds. A similar condition reported in dogs fed generic or low-cost commercial dog foods has been presumed to be related to zinc deficiency secondary to poor zinc bioavailability [102,103]. Some older dogs with poor hair coats and scaly skin may also recover with zinc supplementation [104].

Zinc bioavailability can be influenced by the zinc source, as well as by dietary phytate, supplemental calcium, and other variables. Zinc deficiency in dogs is characterized by scaling, crusting skin lesions, hyperkeratosis, and secondary skin infections responsive to dietary change or zinc supplementation. Elemental zinc (initial dose 2-3 mg/kg/day, divided and given with food) from either zinc sulfate or zinc gluconate has been recommended as an initial dose for dogs with suspected zinc-responsive dermatosis [105,106].

One randomized, double-blinded, placebo-controlled study showed an ameliorative effect of zinc methionine supplementation in dogs with mild to moderate, chronic, non-seasonal AD, receiving either cyclosporine or glucocorticoids [107].

The aforementioned studies provide evidence supporting a potential benefit of adjunctive vitamins and zinc supplementation to prevent nutritional issues and to complement traditional medicine in dermatological problems. Additional studies are needed to extend the use of these compounds as adjunctive treatment for chronic skin disorders in order to allow a reduction of dose or frequency of drug administration.

## 3. Discussion

The studies considered above suggest a possible role for complementary feeds in the management of canine dermatological disorders [16,81,93]. 

Interestingly, in the collective imagination, CFs are seen as a more natural alternative to conventional pharmacological approach [108], with both positive and potentially negative consequences. Moreover, complementary feeds are usually over the counter products, which do not need any prescription, and so are freely available for pet owners, although they are often suggested by the vet. CFs define, hence, a new category, which shades the frontier between drugs and feed, and can be considered as the evolution of the feed supplementation [109]. However, according to European Feed Regulation (Regulation (EC) No 767/2009; No 1831/2003 on the feed materials), most ingredients should be included into feed additives or feed material registers. 

Investigating the clinical effectiveness of feed supplementation in small animals represents a challenge. To obtain reliable data, the studies should be optimized and standardized: blinded randomization, placebo control, and double blinding of both owner and clinician are the minimum requirements for clinical trials aimed to assess the efficacy of complementary compounds. Unfortunately, in literature, only a few studies on that topic with such type of design are available.

Dermatological diseases, especially CAD, need to have filled the lack of knowledge in biological markers to objectively evaluate pathologies and efficacy of a treatment, providing a measurable response of a complementary feed [5]. Besides, the lack of pharmacodynamic and pharmacokinetic data make, in many cases, the dosing an uncertain and mostly empirical process, even with well-known and long-used substances. In this regard, nonappearance of expected effectiveness or possible side effects may be sometimes due to intrinsic hidden factors, which are linked to the plant/food compounds contained in the supplement. Potential undesirable side effects might also derive from extrinsic factors, such as the lack of standardization extraction/production process or standards of good manufacturing practices [110]. Regarding the use of botanical extracts, it is important to consider the type of essence and the extraction method used [67], and its rate of bioavailability, being this last, very often, absolutely poor [111].

The number of active constituents in the plant material, in fact, can be influenced by several environmental factors, as well as by the plant parts and the subspecies used [48]. In this case, in addition to the standardization of the experiment, it is also necessary to standardize the treatment, paying attention, also, at pollutants eventually present as pesticides, herbicides, and polychlorinated biphenyls (PCBs) residue [112]. 

Alongside the paucity of data that veterinarians can rely on, another issue of complementary feeds for dermatological purposes concerns health claims. Accordingly, differentiating between livestock and companion animal regulations may be the most desirable solution to guaranty both safety and efficacy.

## 4. Conclusions

In recent years, the use of complementary feeds in animal health has become more and more popular, and the range of supplements available, alone or included in feed, has tremendously increased. Thanks to improvements in small animal nutrition research, scientific proof of the beneficial effect of some compounds that have a long history of use in human nutrition are becoming accessible, and the use of CFs is very appealing for both pet owners and veterinarians. Filling the gap between regulation, industry, healthcare professionals, and pet owners, and then providing further scientific evidence is a challenge that must be faced in the near future for more conscious use of CFs in veterinary dermatology, and more, in general, in veterinary medicine.

## Figures and Tables

**Table 1 vetsci-07-00038-t001:** Summary of compounds investigated in canine dermatological disorders. See the text for references.

Family of the Parental Compound	Compounds	Effect
Bioactive lipids	*aliamides*	Regulation of cutaneous inflammation and immunity
Polyunsaturated fatty acids	*omega 3 fatty acids* *omega 6 fatty acids*	Shifting of the arachidonic acid cascade towards production of less inflammatory mediators (prostaglandins and leukotrienes)
Botanical extracts	*Rhemannia glutinosa* *Paeonia lactiflora Glycyrrhiza uralensis*	Steroid sparing effect in addition to inflammatory, antioxidant, and antimicrobic properties
	*Arctium lappa* *Althaea officinalis* *Malva sylvestris* *Ribes nigrum*	Antioxidant and anti-inflammatory activities
	*Ascophyllum nodosum* *Cucumis melo* *Haematococcus pluvialis* *Curcuma longa* *Camellia sinensis* *Punica granatum* *Polygonum cuspidatum* *Echinacea purpurea* *Grifola frondosa* *Glycine max*	Reduction of IFNϒ and T lymphocytes, in addition to antioxidant and anti-inflammatory activities
	*Lactobacillus rhamnosus* *Lactobacillus paracasei* K71 *Lactobacillus sakei* *Enterococcus* spp.	Exact mechanism has yet to be fully elucidated; the supposed mechanism is modulation of the immune response toward a Th1 cell-mediated response
Vitamins and Minerals	vitamin A vitamin E vitamin D zinc	Prevention of nutritional deficiencies

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
