# Peer review of "Impact of Nutritional Supplementation on Canine Dermatological Disorders"

_vetsci, 2020, doi:10.3390/vetsci7020038_

Round 1

Reviewer 1 Report

This is, overall, a well-written and comprehensive review of nutritional additives/supplements in veterinary dermatology.  I would suggest changing the title (and some of text) from "complementary feeds" to either nutraceuticals or nutritional additives/supplements.  I have a few suggested areas for clarification/modification:

-Lines 79 and 94 - Is Adelmidrol PEA or another aliamide?  Please clarify.

-Line 118 - I don't think it is true that essential fatty acids are devoid of side effects.  Adverse effects are possible (including altered platelet function and GI effects).  See Lenox CE and Bauer JE. Potential adverse effects of omega-3 fatty acids in dogs and cats. J Vet Intern Med 2013; 27: 217-226.

-Section 2.3 (Botanical extracts) - While I know that the focus of the review is on orally administered therapies, there are several publications about the use of topical plant-derived essential oils in various dermatologic conditions that may be worthwhile discussing (often these are in combination with essential fatty acids).

-Line 165 - This is a bit unclear.  Rewording is suggested.  I think it is also important to keep in mind that seborrhea in dogs is not a true clinical diagnosis, but the result of a number of potential underlying conditions.

-Lines 168-170 - These sentences are redundant with the sentence below.  I suggest removal.

-Line 227 - "...if different topical therapies may have influenced the cutaneous microflora and subsequent interactions with intestinal microflora immune system effects" - the meaning of this portion of the sentence is unclear.  Please clarify.

-Line 281 - This recommended dose of zinc is lower than what is recommended by other authors.  The recommended starting dose of elemental zinc in ZRD is 2-3 mg/kg/day (see White SD, et al. Zinc-responsive dermatosis in dogs: 41 cases and literature review/ Vet Dermatol 2001; 12: 101-109).

-Table 2 - This table doesn't add much to the text since different supplements/therapies may be helpful for each condition.  Additionally, some of the disorders listed are not truly disorders (eg: hyperkeratosis, parakeratosis, seborrhea, chronic pruritus, etc) but clinical or histopathologic features that may have a number of potential underlying causes.  I suggest removal of this table.

-There are also some spelling (eg: line 327 - "guaranty") and grammatical errors throughout the manuscript, though it is overall easy to read.  

Author Response

Dear Reviewer,

Thank you for the revision of the manuscript.

We have revised the paper according to your comments. We used the "Track Changes" function in Microsoft Word, so you can easily pick up changes made.

The manuscript has been checked by a native English-speaking colleague of us to improve the quality of the language.

Below the point-by-point replies to your comments.

This is, overall, a well-written and comprehensive review of nutritional additives/supplements in veterinary dermatology.  I would suggest changing the title (and some of text) from "complementary feeds" to either nutraceuticals or nutritional additives/supplements. 

Thanks for the advice. Title and text changed to reflect this comment. 

I have a few suggested areas for clarification/modification:

-Lines 79 and 94 - Is Adelmidrol PEA or another aliamide?  Please clarify. Yes, adelmidrol is a different aliamide compound. Text modified to reflect this comment

-Line 118 - I don't think it is true that essential fatty acids are devoid of side effects.  Adverse effects are possible (including altered platelet function and GI effects).  See Lenox CE and Bauer JE. Potential adverse effects of omega-3 fatty acids in dogs and cats. J Vet Intern Med 2013; 27: 217-226. Thanks for the comment. We have modified the text and added the reference to reflect this comment.

-Section 2.3 (Botanical extracts) - While I know that the focus of the review is on orally administered therapies, there are several publications about the use of topical plant-derived essential oils in various dermatologic conditions that may be worthwhile discussing (often these are in combination with essential fatty acids). Two sentences have been added at the end of the section 2.3 discussing to reflect this comment

-Line 165 - This is a bit unclear.  Rewording is suggested.  I think it is also important to keep in mind that seborrhea in dogs is not a true clinical diagnosis, but the result of a number of potential underlying conditions. The sentence has been rephrased to clarify and reflect this comment

-Lines 168-170 - These sentences are redundant with the sentence below.  I suggest removal. Sentence removed

-Line 227 - "...if different topical therapies may have influenced the cutaneous microflora and subsequent interactions with intestinal microflora immune system effects" - the meaning of this portion of the sentence is unclear.  Please clarify. Sentence rephrased to clarify.

-Line 281 - This recommended dose of zinc is lower than what is recommended by other authors.  The recommended starting dose of elemental zinc in ZRD is 2-3 mg/kg/day (see White SD, et al. Zinc-responsive dermatosis in dogs: 41 cases and literature review/ Vet Dermatol 2001; 12: 101-109). Thanks for the comment. The text has been modified accordingly to reflect this comment

-Table 2 - This table doesn't add much to the text since different supplements/therapies may be helpful for each condition.  Additionally, some of the disorders listed are not truly disorders (eg: hyperkeratosis, parakeratosis, seborrhea, chronic pruritus, etc) but clinical or histopathologic features that may have a number of potential underlying causes.  I suggest removal of this table.

Table removed

-There are also some spelling (eg: line 327 - "guaranty") and grammatical errors throughout the manuscript, though it is overall easy to read.  Text has been checked by a native English speaking colleague. 

Reviewer 2 Report

check comments on pdf file 

Author Response

Dear Reviewer,

Thank you for the revision of the manuscript.

We have revised the paper according to your comments. We used the "Track Changes" function in Microsoft Word, so you can easily pick up changes made.

The manuscript has been checked by a native English-speaking colleague of us to improve the quality of the language.

Below the point-by-point replies to your comments.

Line 28: term changed to reflect the comment

Line 38: changed to reflect the comment

Lines 83 and 86: terms changed

Line 85: changed

Line 96: changed

Lines 102-103: sentence rephrased to reflect this comment

Line 209: changed

Line 218: changed

Lines 238-240: text rephrased to reflect this comment. A reference was added.

Line 244: changed